# *Punica granatum* (Pomegranate) Peel Extract Pre-Treatment Alleviates Fenpropathrin-Induced Testicular Injury via Suppression of Oxidative Stress and Inflammation in Adult Male Rats

**DOI:** 10.3390/toxics11060504

**Published:** 2023-06-03

**Authors:** Ali B. Jebur, Raghda A. El-Sayed, Mohamed M. Abdel-Daim, Fatma M. El-Demerdash

**Affiliations:** 1Department of Animal Production, College of Agriculture, University of Kerbala, Kerbala 56001, Iraq; blash.ali1966@gmail.com; 2Department of Environmental Studies, Institute of Graduate Studies and Research, Alexandria University, 163 Horreya Avenue, P.O. Box 832, Alexandria 21526, Egypt; raghdaahmed@ymail.com; 3Department of Pharmaceutical Sciences, Pharmacy Program, Batterjee Medical College, P.O. Box 6231, Jeddah 21442, Saudi Arabia; abdeldaim.m@vet.suez.edu.eg; 4Pharmacology Department, Faculty of Veterinary Medicine, Suez Canal University, Ismailia 41522, Egypt

**Keywords:** fenpropathrin, pomegranate peel extract, oxidative stress, molecular/biochemical analysis, testicular dysfunction

## Abstract

Fenpropathrin (FNP) is one of the commonly used insecticides in agriculture and domestically, leading to environmental and health problems. The goal of the current investigation was to determine how well pomegranate peel extract (PGPE) could prevent the testicular toxicity and oxidative stress induced by FNP. Four groups of male Wistar rats were randomly assigned: negative control (corn oil), PGPE (500 mg/kg BW), positive control (FNP; 15 mg/kg BW, 1/15 LD_50_), and PGPE + FNP. For four weeks, the rats received their doses daily and orally via gavage. The major phytochemical components (total phenolic, flavonoids, and tannins contents) detected in PGPE by GC-MS included ellagic acid, hydroxymethylfurfurole, guanosine, and pyrogallol with high total phenolic, flavonoids, and tannin contents. FNP-treated rats showed a marked elevation in testicular levels of thiobarbituric acid-reactive substances, hydrogen peroxide, and protein carbonyl content, as well as the activity of aminotransferases and phosphatases. Meanwhile. a significant decline in body weight, gonadosomatic index, glutathione, protein contents, enzymatic antioxidants, and hydroxysteroid dehydrogenase (3β HSD, and 17β HSD) activity was observed. In addition, significant alterations in testicular P53, Cas-3, Bcl-2, IL-β, IL-10, testosterone, follicle-stimulating and luteinizing hormones, and sperm quality were detected. Furthermore, biochemical and molecular changes were corroborated testicular histological abnormalities. Moreover, PGPE-pretreated FNP-intoxicated rats demonstrated considerable improvement in the majority of the studied parameters, when compared to FNP-treated groups. Conclusively, PGPE provided a potent protective effect against the testicular toxicity caused by FNP, due to its antioxidant-active components.

## 1. Introduction

Pesticides are employed in agriculture to increase food production through the control of disease vectors and the eradication of undesired insects. Since pyrethroids are among the most potent insecticides available and are often used in veterinary care, household use, and pest control, they may have adverse effects on manufacturing employees, field applicators, the environment, and eventually, the general public [1]. Fenpropathrin is a visible light stable pyrethroid insecticide with a broad insecticidal spectrum, long-lasting action, and control effects. It is frequently used on vegetables, fruits, herbal medicine, and other crops in order to combat Lepidoptera, Homoptera, Hemiptera, Coleoptera, and other pests and mites [2,3]. FNP, an alpha-cyano pyrethroid, alters the voltage-sensitive sodium channels’ gating kinetics, disrupting neuronal function and resulting in immediate neurotoxic effects in both insects and other organisms as the main mechanism of action [4,5,6]. Pyrethroid insecticides cause a number of harmful effects that most likely result from oxidative stress, the generation of reactive oxygen species (ROS), and reactive metabolites damaging various cell membrane components [7,8]. It is well established that several pesticides now in use negatively affect male reproductive capability in lab, field, clinical, and professional contexts [9]. According to published studies, pyrethroids can disrupt the homeostasis of oxidative stress, alter the antioxidant defense mechanism [9], and cause testicular degeneration, male reproductive failure, and malformations in the fetus of rodents after repeated exposure [10], as well as upregulating the expression of enzymatic antioxidants in mouse liver [11].

Pomegranate (*Punica granatum* L.) is a plant that is frequently used in both food and traditional medicine in China [12]. Its height ranges from 5 to 10 m, and it belongs to the family Lythraceae, subfamily Punicoideae, kingdom Plantae, order Myrtales, and genus Punica. Pomegranate contains abundant phytochemicals that are potent antioxidants, including polyphenols, flavonoids, anthocyanins, tannins, ellagic acid, gallic acid, and catechins [13,14]. These components act by enhancing oxidative biomarkers, scavenging reactive oxygen species, and neutralizing them [15]. Numerous studies have shown that extracts from different parts of the pomegranate plant have a wide range of biological effects, including antibacterial, antiviral, antioxidant, cancer prevention, anti-inflammatory, cardioprotective, and anti-diabetic properties, as well as improvement in sperm quality [16,17,18,19,20]. When compared to other fruits, the pomegranate fruit exhibits strong antioxidant activity and high polyphenol content [21,22], which can protect against the harm caused by carbon tetrachloride and lead during spermatogenesis [23,24]. Moreover, treatment with pomegranate peel and seed oil extracts enhanced liver functioning, reduced DNA fragmentation, caspase-3, and malondialdehyde levels, and increased enzymatic and nonenzymatic antioxidants due to diethylnitrosamine injection [25]. The objective of the present investigation was to evaluate the potential antioxidant roles of pomegranate peel extract against oxidative stress, changes in antioxidant status, perturbations in biochemical and molecular indices, and testicular dysfunction induced by FNP.

## 2. Materials and Methods

### 2.1. Materials

*Punica granatum* (Pomegranate) was bought at a local market in Alexandria, Egypt, recognized, and authenticated by the Botany Department, Faculty of Science, Alexandria University. Fenpropathrin (>92% purity) was supplied by Zhejiang Dongyang-Xinhui Chemical Industry Co., LTD., Jinhua, Zhejiang, China, CAS 39515-41-8.

### 2.2. Punica granatum Peel Extract Preparation

Briefly, the peels were physically separated, dried in the sun for 36 h, then ground into a powder using a grinder with 60 mesh sizes. Using a magnetic stirrer and 1000 mL of 80% ethanol, the peel powder (100 g) was extracted for 48 h at 25 °C and then filtered using Whatman No. 41 filter paper to remove peel remnants. The residue was separated with 500 mL of ethanol, filtered, and the extracts were placed in an oven at 37 °C for dryness for 30 h, and kept at 4 °C until utilized [13].

### 2.3. Gas Chromatography/Mass Spectrometry(GC/MS) Analysis of Punica granatum Peel Extract

The chemical components of the extract were identified using Thermo Scientific GC/MS version (5) 2009 equipment, with a TG-5MS column (30mX0.32mmID) [26]. The components of the extract were determined using mass fragmentation patterns and the National Institute of Standards and Technology’s (NIST) mass spectrum database (version 2).

### 2.4. In Vitro Measurement of Total Phenolic, Flavonoids, and Tannins Content

The total phenolic contents (TPC) of *P. granatum* peel extract were analyzed using the technique of Parsaei et al. [27], and the absorbance of the mixture was measured at a wavelength of 725 nm. The results were reported in terms of the gallic acid equivalent of dry extract (mg GAE/g). The Yang et al. [28] technique was used to determine the total flavonoid content (TFC), the absorbance was detected at 510 nm, and the results were represented as the mg catechin equivalent (mg CAE/g) per gram of dry extract. Hydrolysable tannins were determined using the method of Willis and Allen [29], and measured at a wavelength of 580 nm. The results were expressed as mg tannic acid equivalent per g of dry extract (mg TAE/g).

### 2.5. Experimental Design

Twenty-eight male Wistar rats (160–170 g) were supplied by the Faculty of Medicine, Alexandria University, Alexandria, Egypt. The animals were acclimated for two weeks in stainless steel cages, with seven animals per cage, a commercial meal, and unlimited access to tap water (temperature: 21 °C; photoperiod: 7 a.m. to 7 p.m.) before the beginning of the treatment. The ethical local committee of Alexandria University, Alexandria, Egypt, approved the experimental design, which also follows the National Institutes of Health Guidelines. The rats were divided into four groups: group 1 served as a negative control group, and received corn oil; group 2 received *P. granatum* peel extract (PGPE; 500 mg/kg BW); group 3 (positive control) received fenpropathrin dissolved in corn oil (FNP; 15 mg FNP/kg BW; 1/15 of LD_50_); and group 4 received PGPE one hour prior to the administration of FNP. *P. granatum* peel extract and FNP dosages were given orally by gavage for four weeks, daily, according to the method of Ahmed and Zaki [30] and Xiong et al. [31], respectively. The FNP dosage was chosen based on the previous preliminary experiment of [31], during which the authors studied the effects of administration of 7.5, 11.25, 15 and 22 mg/kg/day, which is equivalent to 1/30, 1/20, 1/15, and 1/10 of LD_50_ for male rats. During this preliminary study, the authors found that 22 mg/kg/day FNP was lethal to most of the rats within 2 days of exposure, while the lowest dose of 7.5 mg/kg/day did not induce any mortalities in or change the behavior of the tested rats after 1 month of exposure. Thus, we chose 15 mg/kg/day as the experimental dose for adult male rats. At the end of the experimental period, the animals were fasted overnight (12 h), weighed, and given isoflurane anesthesia, and their blood and testicles were harvested. Three sections of the testes were created: for histology, a section was fixed in 10% formalin; for biochemical research, a section was stored at −20 °C; and for molecular analysis, a section was kept at −80 °C.

### 2.6. Blood Collection and Serum Sample Preparation

Heart puncture blood samples were taken, and they were allowed to stand for half an hour at 25 °C before centrifuging at 3000× *g* for 15 min. The clear serum was removed and stored at −20 °C. 

### 2.7. Evaluation of Sperm Quality

The left caudal portion of each testicle’s epididymis was gently removed before being minced in 5 mL of Hanks’ buffered salt solution and left at room temperature for 15 min, to allow spermatozoa to move into the fluid. According to Adamkovicova et al. [32], the sperm parameters were evaluated using a bright field microscope (Olympus, Tokyo, Japan) and computer-assisted semen analysis. The sperm samples were diluted with physiological solution (10 μL), pipetted into a Makler Counting Chamber, and immediately assessed. Within each of the measurements made with the CASA system, parameters from a minimum of seven fields of the Makler Counting Chamber were analyzed, and 1000 sperm were evaluated per sample.

### 2.8. Hormone Analyses

The testosterone concentration was assessed utilizing a radioimmunoassay kit (RIA TESTO CTC KIT) provided by Dia-Sorin Company: Stillwater, MI, USA, while the luteinizing hormone (LH) level was evaluated using RIA kits acquired from NIADDK, Bethesda, MD, USA. Immunodiagnostic tools and the Elisa Kit test were used to assess the serum level of follicle-stimulating hormone (FSH) (DiaMetra kits, Via Giustozzi, Spello, Italy).

### 2.9. Testis Homogenate Preparation

The testes were separated from the dissected rats and homogenized (10% *w*/*v*) in 1.15% KCl, an ice-cold potassium phosphate buffer (0.01 mol/L, pH 7.4). After centrifuging the homogenates at 10,000× *g* (4 °C) for 20 min, the supernatants were collected and kept at −20 °C until they were employed in subsequent tests. 

### 2.10. Determination of Oxidative Stress, Non-Enzymatic and Enzymatic Antioxidants 

The thiobarbituric acid-reactive substances (TBARS), hydrogen peroxide (H_2_O_2_), and protein carbonyl content (PCC) were estimated in the testes homogenate, according to Ohkawa et al. [33], Velikova et al. [34], and Reznick and Packer [35], respectively. The reduced glutathione (GSH) content was determined using Ellman’s method [36]. The activity of superoxide dismutase (SOD; EC 1.15.1.1) and catalase (CAT; EC 1.11.1.6) was assessed using the methods of Misra and Fridovich [37] and Aebi [38], respectively. The activity of glutathione peroxidase (GPx; EC 1.11.1.9) and glutathione reductase (GR; EC 1.6.4.2) was evaluated using the method of Hafeman et al. [39], while glutathione *S*-transferase (GST; EC 2.5.1.18)’s activity was determined according to Habig et al. [40].

### 2.11. Determination of Testicular Enzymes, Aminotransferases, Phosphatase Activity, and Protein Content

The activities of testicular 3β hydroxysteroid dehydrogenase (3β-HSD; EC 1.1.1.145) and 17β hydroxysteroid dehydrogenase (17β-HSD, EC 1.1.1.51) were assessed according to Talalay [41]. Aspartate aminotransferase (AST; EC 2.6.1.1, CAT.NO. AS 10 61 (45)), alanine aminotransferase (ALT; EC 2.6.1.2, CAT.NO. AL 10 31 (45)), acid phosphatase (ACP; EC 3.1.3.2, CAT. NO. AC 10 10), and alkaline phosphatase (ALP; EC 3.1.3.1, CAT. NO. AP 10 20)’s activities and protein content (CAT. NO. TP 20 20) were determined using commercial kits (Biodiagnostic, Giza Egypt), and the absorbance was measured spectrophotometrically (Jenway^®^ Model 7315 UV/Visible spectrophotometer, Evreux Cedex, France).

### 2.12. Quantitative Real-Time PCR 

By utilizing the Qiagen tissue extraction Kit (Qiagen, USA) and following the instructions of the manufacturer, the total RNA was extracted from testis tissues. The concentration and purity of RNA were determined using a NanoPhotometer^®^ spectrophotometer (IMPLEN, St. Louis, CA, USA). Following the instructions given by the manufacturer, a high-capacity cDNA reverse transcription Kit (Fermentas, Waltham, MA, USA) was utilized to convert total RNA (0.5–2 µg) into cDNA. Real-time qPCR analysis and amplification were carried out using an Applied Biosystem with software version 3.1 (StepOneTM, USA). At the annealing temperature, the qPCR assay was optimized with the primer sets. The β-actin gene was utilized as a reference (internal control) to calculate the fold change in target genes. The relative quantification was estimated using the Applied Biosystem 7300/7500/7500 Fast software, according to the 2^−∆∆Ct^ method. The RQ is the fold change compared to the calibrator (untreated sample). Table 1 displays the sequence of the primers [42].

### 2.13. Measurement of Inflammatory and Anti-Inflammatory Cytokines

Interleukin 10 (IL10) (Cat. No: 201-11-0109) and Interleukin 1β (IL1β) (Cat. No: 201-11-0120) ELISA kits (Sigma Aldrich^®^, St. Louis, MO, USA) were utilized to evaluate IL10 and IL1β protein levels in the rat testes, following the instructions of the manufacturer.

### 2.14. Histological Investigation

Testes were fixed in 10% formalin solution, and serial paraffin sections (5 µm) were prepared to analyze the histological alterations using hematoxylin and eosin staining [43]. Following that, slides were examined and photographed using a light microscope (Olympus BX 41Orinpasu Kabushiki-gaisha, Tokyo, Japan).

### 2.15. Statistical Analysis

Data from different groups were represented as means ± standard errors of the mean (SEM), and then SPSS software was used to analyze them (version 22, IBM Co., Armonk, NY, USA). Through the use of ANOVA and Tukey’s post hoc test, groups were compared. At *P* ≤ 0.05, significant values were accepted.

## 3. Results

### 3.1. GC-MS and Phytochemical Analysis

GC-MS analysis of *P. granatum* peel ethanolic extract indicated the presence of thirty peaks for different phytochemical components. They are described and identified by comparing the mass spectra of the constituents to those of the phytoconstituents. Ellagic acid, hydroxymethylfurfurole, guanosine, and pyrogallol are the main phytochemical constituents’ whose mass spectra were observed (Figure 1). Additionally, the concentrations of the standards gallic acid, catechin and tannic acid were derived from the following standard curves ranging from 0 to 400 mg/g, y = 0.0412x + 0.026, (R^2^ = 0.974) y = 0.0016x − 0.004 (R^2^ = 0.9518), and y = 0.0412x + 0.026, (R^2^ = 0.974), respectively, where y = %R at the steady state and x = amount of standard used in the reaction. The results showed that the total phenolic content of pomegranate peel extract was substantial, with a value of 211 mg GAE/g DW. Additionally, this peel extract has total tannins of 106 mg TAE/g DW, and total flavonoids of 59.84 mg/g CAE (Table 2 and Figure 2).

### 3.2. Body and Organ Weights

Animals from different groups appeared to be normal throughout the experiment, with the exception of the increased salivation in the FNP-treated group. The results demonstrated that administration of FNP considerably reduced the final body and testes weights and gonadosomatic index when compared to the control group. Furthermore, rats given PGPE and then FNP revealed a significant improvement in these parameters as compared to the FNP-exposed group. Intake of PGPE alone had no discernible impact (Table 3).

### 3.3. Sperm Parameters and Hormone Levels

Animals administered with FNP had a significantly (*P* < 0.05) lower level of sperm characteristics, including normality, motility, viability, and count, when compared to control rats. Rats pre-administered with PGPE followed by FNP treatment displayed significant improvement in sperm quality in comparison to the FNP-intoxicated group. Furthermore, the results revealed significant alterations in the concentration of testosterone, LH, and FSH in the rat serum of the FNP-treated animals when compared to the control levels. Rats administered with PGPE and then treated with FNP exhibited considerable improvement in the concentration of hormones when compared to the FNP group. Supplementation of PGPE alone did not affect the studied sperm quality and hormones compared to the control group (Figure 3).

### 3.4. Oxidant Stress and Antioxidant Biomarkers 

The levels of TBARS, H_2_O_2_, and PCC were considerably increased, while the activities of SOD, CAT, GPx, GR, and GST, as well as GSH content, were reduced significantly (*P* < 0.05) in the testes homogenate of rats treated with FNP as compared to the control rats. Additionally, animals supplied with PGPE and later given FNP showed significant amelioration in the measured parameters, in comparison to rats given FNP alone. Furthermore, rats given PGPE alone revealed considerable improvement in the studied biomarkers when compared to the control rats (Table 4).

### 3.5. Aminotransferases, Phosphatases, 3 β, and 17 β Hydroxysteroid Dehydrogenase Activities and Protein Content

The results indicated that the activities of ALP, ACP, AST, and ALT in rat serum and testes protein content as well as 3β-HSD and 17β-HSD activities in rat testis homogenate were significantly (*P* < 0.05) altered in rats treated with FNP, compared to the control group. Furthermore, animals that received PGPE and then were given FNP showed an important amendment in the studied parameters, compared to the FNP group. PGPE supplementation by itself had no discernible effects on the parameters under investigation (Table 5).

### 3.6. Gene Expression, Inflammatory and Anti-Inflammatory Cytokines

Results revealed a significant (*P* < 0.05) upregulation of p53 and Cas3 expression, and a downregulation of Bcl-2 expression. In addition, IL-1 and IL-10 levels were significantly altered in FNP-intoxicated rats when compared to the control rats. Compared to the FNP group, rats given PGPE and afterward treated with FNP showed a significant modification in the assessed parameters (p53, Cas3, Bcl-2, IL-1, and IL-10) (Figure 4).

### 3.7. Histological Observation of Testis

Testicular sections from controls (G1) and PGPE (G2) that underwent histopathological investigation revealed normal testes architecture, full spermatogenesis, and a high sperm concentration in the seminiferous tubules lumen. On the contrary, spermatogenesis-differentiated and wild lumen were reduced, and there were noticeable damaged seminiferous tubules in the testicles of FNP (G3). The destructive epithelial lining of the seminiferous tubules, numerous hyperchromatic spermatogenesis, loss of their differentiated spermatogenesis, necrotic spermatozoa with wide lumens, markedly dilated interstitial tissues with atrophied Leydig cells, and seminiferous tubules were observed. Rats that received both PGPE and FNP (G4) showed improvement in their seminiferous tubule structure, with regular spermatogenesis (Figure 5 and Table 6).

## 4. Discussion

Endocrine disruption and testicular dysfunction are side effects of environmental toxins that have a negative impact on both people and wildlife [44]. Human male fertility was decreased as a result of these pollutants’ imitation of natural estrogens and their effects on spermatogenesis, steroidogenesis, and the activity of Leydig and Sertoli cells [45]. The current study investigated how PGPE can lessen the oxidative stress and testicular toxicity caused by FNP. The ability of FNP to enhance activity and excessive salivation, appetite loss, diarrhea, and occasionally vomiting may be responsible for the observed drop in final body weight (Table 3), which is an indicator of toxicity, because organism fitness is reduced [9]. Additionally, the observed decrease in sex hormones, particularly testosterone, may be responsible for the alteration in testis weight and the gonad somatic index, and for the declining semen parameters [46]. In agreement, FNP intoxication affects testes’ weight through the suppression of spermatogenesis, steroidogenic enzymes, and the reduction of germ cells, which in turn increases the abnormal sperm percentage [9]. Additionally, it was found that FNP induces cell death as a result of ROS generation, which may help to explain the dramatic changes in sperm quality seen in rats receiving FNP treatment (Figure 3) [31].

Male reproductive functions, such as the generation of androgens and spermatozoa, are initiated and maintained by hormones [47]. The ability of the testes to metabolize xenobiotics is said to be critical in the development of testicular toxicity [48]. Otherwise, pyrethroid-induced oxidative stress is brought on by mitochondrial dysfunction and a decrease in the activity of mitochondria complex I (as cytochrome P450), which serves as an oxidant for various processes of androgen synthesis [49,50]. In addition, changes in hormone levels and sperm characteristics in pesticide-exposed rats were discovered to have detrimental impacts on fertility and seminiferous tubule destruction [51,52]. Moreover, the two important dehydrogenases, 3β-HSD and 17β-HSD, control testicular steroidogenesis; they are directly engaged in the production of androstenedione and testosterone from pregnenolone. Furthermore, the observed fall in 3β-HSD and 17β-HSD activity (Table 5) may lead to a reduction in steroidogenesis in rats, as the production of the majority of biologically active steroids involves these enzymes [53].

Pyrethroids have been demonstrated to damage cells via oxidation, and to accumulate peroxidation products, which are employed as indicators of oxidative stress [8,54]. The testes are frequently exposed to low oxygen levels as a result of their constrained vascularity. However, the testes are particularly susceptible to oxidative and peroxidative damage because they have a high concentration of unsaturated fatty acids [55]. The harmful effects produced by FNP damage cells by limiting membrane mobility, in addition to their lipophilicity, which makes it easy to permeate through the cellular membrane. This may explain the observed increase in TBARS, H_2_O_2_, and PCC (Table 4) [9,54,56].

Endogenous antioxidant mechanisms, both enzymatic and non-enzymatic, work together to defend cells from possible oxidative damage brought on by xenobiotics [52]. In the current investigation, FNP-treated rats displayed a decline in the level of GSH as well as the activity of SOD, CAT, GPx, GR, and GST (Table 4). These outcomes are attributable to oxidative stress, ROS production, or a deficiency in antioxidant defense mechanisms [52]. Since glutathione is a crucial non-enzymatic antioxidant included in protecting cell integrity, its reduction heralds the progression of free radical-induced cell injury. GSH’s sulfhydryl group is oxidized during metabolic activity, resulting in the creation of the equivalent disulfide molecule, oxidized glutathione (GSSG) [57]. As part of the defense mechanism against oxidative damage, superoxide dismutase spurs the diversion of superoxide anion to O_2_ and H_2_O_2_, which is then reduced by a catalase enzyme to H_2_O. The increased ROS production by FNP may be to blame for the decline in both SOD and CAT activity. GPx shields membrane lipids from oxidative degradation by accelerating the reaction between hydroperoxides with GSH to produce GSSG, while GST plays a crucial role in converting xenobiotics into nontoxic metabolites [54]. Moreover, GR catalyzes the redox cycling of GSSG, while glucose-6-phosphate dehydrogenase supplies the necessary reducing agent, NADPH, for this process [58]. Consequently, antioxidant enzymes can reduce and restrict the damaging effects that ROS can have on cells.

Lipid peroxidation (LPO) may be responsible for the detected alterations in enzyme activities (alanine aminotransferase; ALT, aspartate aminotransferase; AST, alkaline phosphatase; ALP, and acid phosphatase; ACP) in rats exposed to FNP because it compromises the integrity of cell membranes and causes cytoplasmic enzymes to seep into the bloodstream [59]. The observed increase in aminotransferases (AST and ALT) due to FNP administration may be brought on by cell injury, changes to the cell membrane’s permeability, and adjustments to protein synthesis or catabolism [60]. Alkaline and acid phosphatase enzymes are crucial in the detoxification, metabolism, and biosynthesis of energy macromolecules. In the present study, FNP administration caused changes in ALP and ACP activity, which is in agreement with the work of Rahman et al. [61]. Acid phosphatase enzymes are essential for a variety of functions such as “cell metabolism, autolysis, differentiation, and blood capillaries’ dilatation in between seminiferous tubules” [62]. It is possible that cellular necrosis, which causes enzyme seepage into the bloodstream, is to blame for the drop in ALP activity in the testes [61]. Moreover, protein depletion is associated with loss of cell function and dysfunction in protein metabolism, since protein is one of the cell components most susceptible to free radical damage; this is consistent with the results of the current study (Table 5) [52].

Pro-inflammatory cytokines such as interleukin-10 (IL-10), tumor necrosis factor-α (TNF-α), and interleukin-1β (IL-1β) are frequently produced in excess in conjunction with the inflammatory response [63]. Various apoptotic molecules, such as cytochrome c and p53, are linked to apoptotic signaling processes [64]. P53 activation and the release of cytochrome c from the mitochondria into the cytoplasm are strong indicators of an apoptotic state in cells [65]. According to the findings in this study (Figure 4), FNP may trigger apoptosis in testis tissues via activating a number of apoptotic-regulated genes, including the highly upregulated p53 gene. In contrast, expression of the anti-apoptotic gene Bcl-2 was downregulated. The high expression of the p53 pro-apoptotic gene that was detected in the FNP-treated group may be the result of increased Cas3 upregulation brought on by increased cell death, itself induced by the disruption of the energy utilization pathway due to FNP’s toxic influence on ATP. Studies have demonstrated a clear relationship between higher cleaved Cas3 expression and apoptosis in many organisms [66]. Moreover, the testes of rats treated with FNP have various abnormalities, according to histopathological examinations (Table 6 and Figure 5). This is due to the fact that FNP damages the testes through oxidative processes, which may have accelerated germ cell death, decreased sperm count, and compromised the integrity and function of the gonads. In agreement, several authors have reported similar histopathological findings in rats exposed to synthetic pyrethroids [9,31,67].

Pomegranate peel is abundant in polyphenols, ellagic acid, ellagitannins, and tannins, all of which have strong antioxidant activity and are used in the treatment of male rat infertility [68]. This suggests that the peel also functions as a metal chelator, an inducer of enzymes that fight oxidative stress, and an electron donor in scavenging free radicals [69]. The pharmacological effects of flavonoids, which are significant and active components of *P. granatum* peel extract, are related to their antioxidant activity, ability to scavenge OH^•^ and O_2_^•^, ability to chelate metal ions, and capacity to collaborate with other antioxidant metabolites [70,71,72,73]. Additionally, tannins may exert their anti-inflammatory effect through the inhibition of enzyme activity, precipitation of membrane proteins, and depletion of metal ions [18]. Phytochemical analysis revealed that pomegranate peel extract tested positively for tannins, flavonoids, and phenolics, which have potent antioxidant effects [74,75]. Pomegranate supplementation boosted antioxidant enzyme activity while decreasing the high amounts of lipid peroxidation byproducts. These enhancements suggest that pomegranate, because of its high antioxidant content, can help to reduce oxidative stress [76,77].

The observed modification in oxidative stress markers and antioxidant status in the testes of rats treated with PGPE and FNP may be due to the phenolic components in PGPE, which have a variety of biological activities, including removing free radicals, inhibiting oxidants, and microbial growth [78], as well as reflecting its immense antioxidant properties [79]. Additionally, PGPE can counteract the harmful effects of ROS by scavenging reactive metabolites, either by halting their formation or reducing their onslaught [80]. Furthermore, pomegranate peel extract contains ellagic acid, a naturally occurring phenolic component with a polyphenolic structure that suppresses the generation of lipid peroxide and exhibits DPPH-free radical scavenging activity [81]. In agreement with the current investigation, the administration of pomegranate fruit and peel extracts has been demonstrated to reduce lipid peroxidation, increase GSH levels, and improve CAT activity in the liver, kidney, and heart [76,77,82]. The glutathione redox cycle is therefore used to protect cellular components from oxidation, maintain cell integrity, and detoxify ROS as a result of the increased GSH level.

As PGPE could enhance the secretion of sexual hormones and antioxidants, and modulate apoptotic gene expression as well as the weights of the animals’ bodies and testicles in rats with relative sterility, its administration decreased the adverse effects of FEN on spermatogenesis. Additionally, a previous study demonstrated that ellagic acid treatment partially reversed the effects of cisplatin on the testes of male rats, and worsened histopathologic results [16]. Interestingly, pomegranate has been discovered to have androgenic properties that are beneficial for addressing issues related to male infertility, including improving sperm quality [83]. Moreover, similar studies have demonstrated that ellagic acid could stop the shrinkage of spermatogonia, Leydig, and Sertoli cells, as well as the diameter of spermatozoa tubules in the testicular tissues of Matern Newborn Rats exposed to cadmium chloride [84]. Furthermore, ellagic acid markedly boosted spermatogenic lineage, which resulted in a marked rise in the diameter and thickness of the seminiferous tubule epithelium [85]. So, *P. granatum* peel is crucial in reducing the oxidative stress that keeps spermatogenesis normal and preserves spermatogenic cells.

Additionally, intake of PGPE prior to FNP intoxication modulated the relative expression of P53, Caspase-3, Bcl-2, and cytokine levels (IL1β and IL10), as the peel extract may inhibit ribonucleic acid (RNA) replication [86]. Punicalagin, one of the phytoconstituents of *P. granatum* peel, has been proven to regulate cell death, particularly apoptosis and pyroptosis. Treatment with punicalagin decreases Bax (pro-apoptotic factor) and upregulates Bcl-2 (anti-apoptotic factor), which leads to the downregulation of caspases (caspase 3, 8, and 9) involved in apoptotic cell death [87]. In accordance, Sayed et al. [88] demonstrated that pomegranate peel’s antioxidant activities decrease oxidative damage by reducing the levels of IL-1β in rat testis. Furthermore, daily consumption of PGPE resulted in a significant increase in spermatogenic cell concentration, which could be due to a reduction in ROS parameters. PGPE possesses strong antioxidant properties due to its high content of polyphenols, particularly ellagitannins (EA), which can easily pass through the mitochondrial membrane [89,90,91]. In conclusion, PGPE can successfully prevent testicular toxicity brought on by FNP, and this property may be attributed to its antioxidant and chelating capabilities.

## 5. Conclusions

The current finding indicates that FNP has the potential to produce testicular dysfunction through oxidative injury; changes in antioxidant defense status, sperm quality, hormones, and tissue architecture; and alterations in biochemical and molecular indicators. Moreover, PGPE was found to contain numerous potent phytochemical components with strong anti-inflammatory and antioxidant capabilities that can relieve FNP toxicity by suppressing oxidative stress and inflammation in rat testes. Therefore, PGPE may be used as an adjuvant therapy for testicular toxicity.

## Figures and Tables

**Figure 1 toxics-11-00504-f001:**
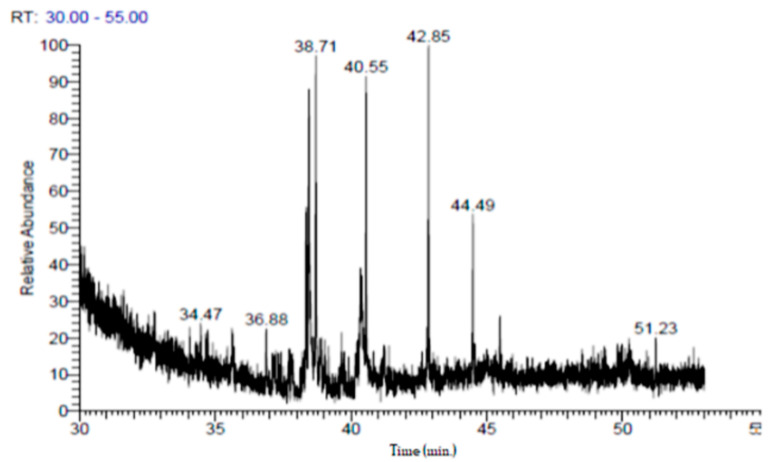
Representative GC-MS chromatogram of *P. granatum* peel extract and the fragmentation pattern of its constituents (retention time 30 min to 55 min).

**Figure 2 toxics-11-00504-f002:**
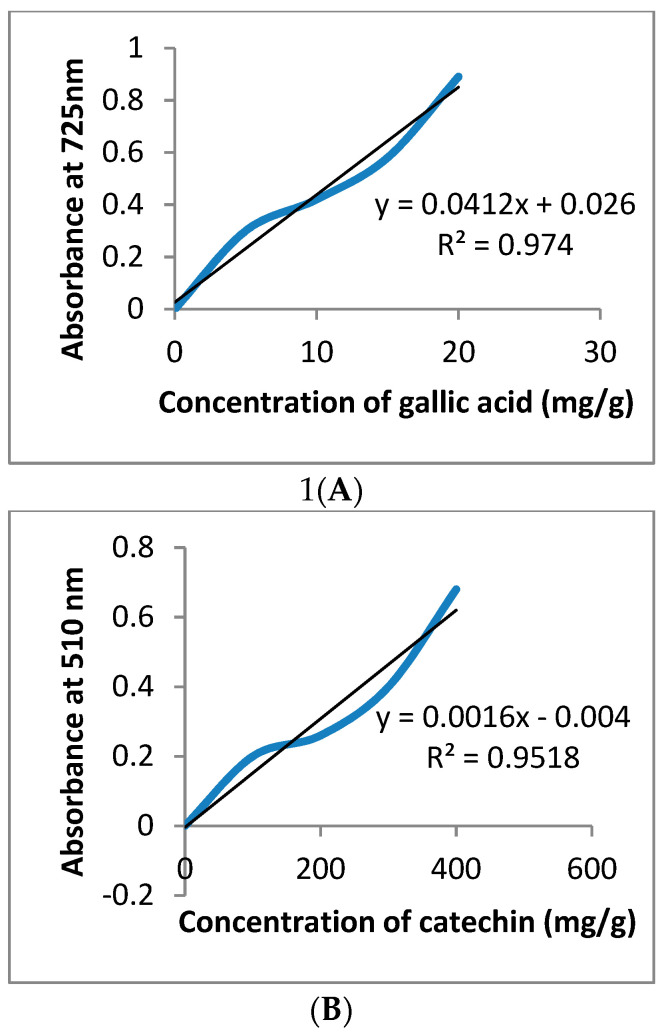
Calibration curve of gallic acid (**A**), to determine the total phenol content spectrophotometrically with different concentrations (0 to 20); catechin (**B**), to determine total flavonoids spectrophotometrically with different concentrations (0 to 400); and tannic acid (**C**), to determine total tannins spectrophotometrically with different concentrations (0 to 20). Each point represents the mean of three experiments.

**Figure 3 toxics-11-00504-f003:**
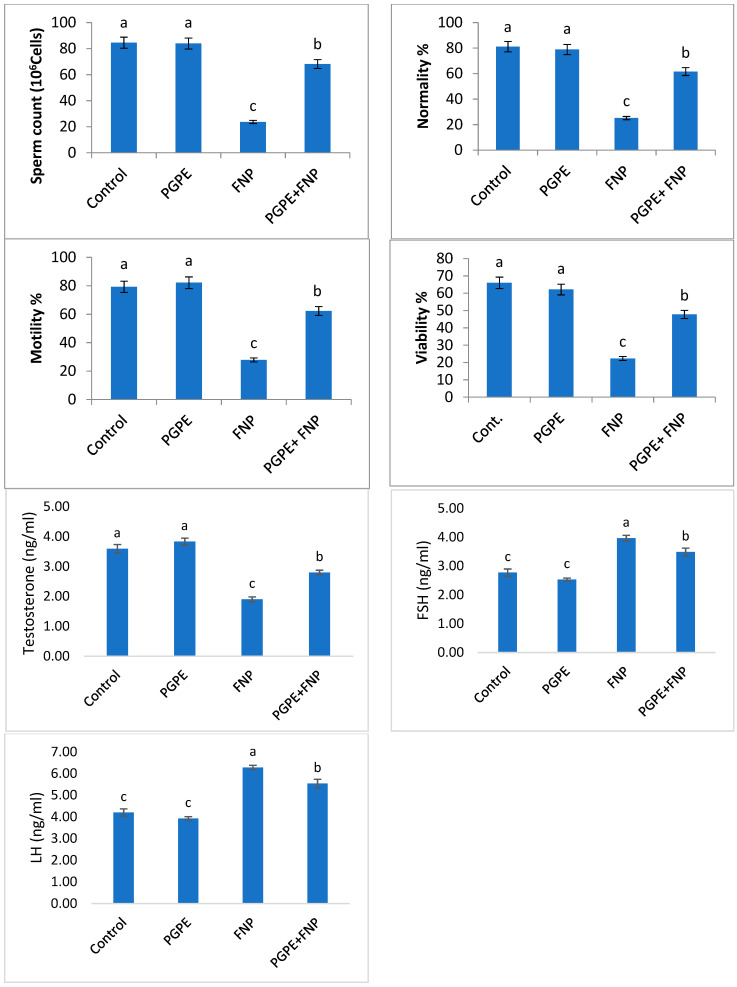
Sperm quality and hormone levels in rats of different groups. Values are expressed as means ± SEM; *n* = 7/group. Significance (*P* < 0.05) between groups was indicated in each row with different superscript letters (a,b,c). Statistically significant variations are compared as follows: PGPE and FNP groups are compared to the negative control group, while the PGPE + FNP group is compared to the positive control group, FNP. Abbreviations: PGPE = *Punica granatum* peel extract, FNP = fenpropathrin, PGPE + FNP = *Punica granatum* peel extract + fenpropathrin. FSH = follicle-stimulating hormone, LH = luteinizing hormone.

**Figure 4 toxics-11-00504-f004:**
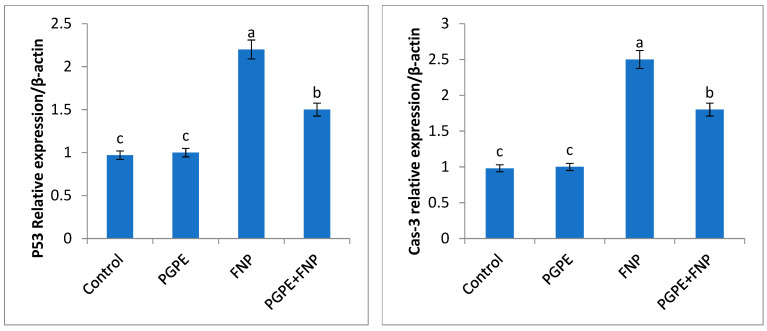
Relative expression of p53, Caspase-3, Bcl-2, and cytokine levels (IL1β and IL10) in the testes tissues of rats of different groups. Data are presented as fold change (mean) ± SEM (*n* = 7/group). Different letters (a,b,c,d) show significant differences at *P* < 0.05. Statistically significant variations are compared as follows: the PGPE and FNP groups are compared to the negative control group, while the PGPE + FNP group is compared to the positive control group, FNP. Abbreviations: PGPE = *Punica granatum* peel extract, FNP = fenpropathrin, PGPE + FNP = *Punica granatum* peel extract + fenpropathrin.

**Figure 5 toxics-11-00504-f005:**
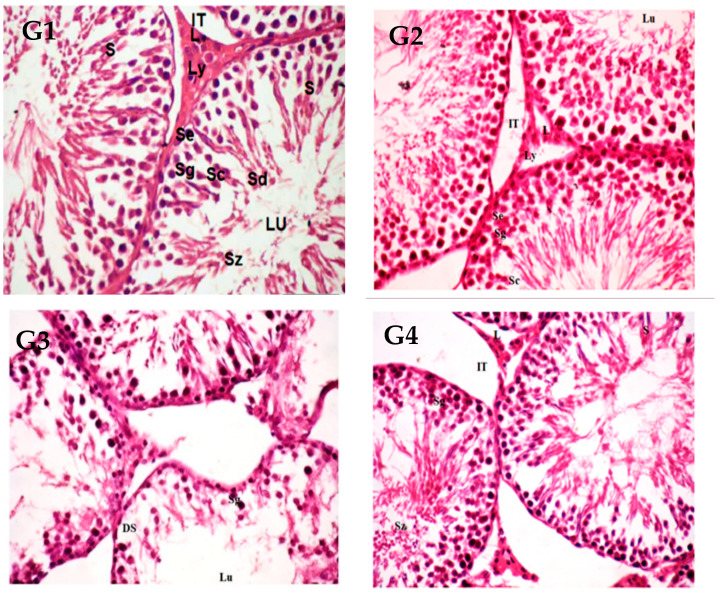
Rat testes sections in control (G1) and PGPE (G2) groups showed a normal structure of seminiferous tubules (S) lined by spermatogenic cells at different stages of maturation; spermatogonia (Sg), spermatocytes (Sc), spermatid (Sd) and free spermatozoa (Sz) filled the lumen (LU). The interstitial tissue (IT) contains Leydig cells (Ly), lymphocytes (L), and mildly dilated blood vessels (BV). FNP group (G3) showed a destructive epithelial lining of the seminiferous tubules (ST); many hyperchromatic spermatogeneses (Sg) lost their differentiated spermatogenesis (DS). There were necrotic spermatozoa with a wide lumen (L), and marked dilated interstitial tissue with atrophic Leydig cells. There is a notable atrophy tubule (AT). The PGPE + FNP group (G4) showed a moderate improvement in the irregular basement membrane of seminiferous tubules, with recovered spermatogonia cells (Sg) in differentiated stages. The seminiferous lumen (L) has many spermatozoa (Sz), markedly dilated interstitial tissues (IT), dilated blood vessels (BV), and numerous lymphocytes (L) (H&E stain ×400; Scale bar 50 µm). Abbreviations: PGPE = *Punica granatum* peel extract, FNP = fenpropathrin, PGPE + FNP = *Punica granatum* peel extract + fenpropathrin.

**Table 1 toxics-11-00504-t001:** Forward and reverse primers used in the qRT-PCR.

Gene	Forward Primer	Reverse Primer
P53	AACTGGAAGAATTCGCG GCCGCAGGAAT	GCTACCCGAAGACCA AGAAGG
Caspase3	GGTATTGAGACAGAC AGTGG	CATGGGATCTGTTTC TTTGC
Bcl-2	ATCGCTCTGTG GATGACTGAGTAC	AGAGACAGCCAGGAGA AATCAAAC
β-actin	AAGTCCCTCACCCTCCCAAAAG	AAGCAATGCTGTCACCTTCCC

**Table 2 toxics-11-00504-t002:** Phytochemical components of total phenolic contents, total flavonoids, and total tannins in *Punica granatum* peel extract (PGPE).

Parameter	PGPE
Total phenolic contents	211.2 ± 5.61 (mg GAE/g DW)
Total flavonoid	59.84 ± 6.41 (mg/g CAE)
Total tannins	106 ± 11.35 (mg/g TAE)

Values represent three replicates.

**Table 3 toxics-11-00504-t003:** Body and testes weights and gonadosomatic index in male rats treated with *Punica granatum* peel extract (PGPE), fenpropathrin (FNP), *Punica granatum* peel extract + fenpropathrin (PGPE + FNP).

Parameters	Groups			
	Control	PGPE	FNP	PGPE + FNP
Initial body weight (g)	164.71 ± 0.86 ^a^	166.68 ± 0.60 ^a^	167.00 ± 2.31 ^a^	163.58 ± 1.89 ^a^
Final body weight (g)	190.33 ± 0.33 ^a^	188.33 ± 0.33 ^a^	144.20 ± 1.53 ^c^	174.89 ± 1.62 ^b^
Body weight gain (g)	25.62 ± 0.49 ^a^	21.65 ± 0.31 ^a^	−22.8 ± 0.87 ^c^	11.3 ± 0.31 ^b^
Testes weight (g)	1.63 ± 0.05 ^a^	1.70 ± 0.06 ^a^	0.82 ± 0.07 ^c^	1.17 ± 0.08 ^b^
Gonadosomatic index	0.85 ± 0.03 ^a^	0.90 ± 0.06 ^a^	0.56 ± 0.05 ^c^	0.66 ± 0.051 ^b^

Values are expressed as means ± SEM; *n* = 7/group. Significance (*P* < 0.05) between groups is indicated in each row with different superscript letters (a,b,c). Statistically significant variations are compared as follows: PGPE and FNP groups are compared to the negative control group, while the PGPE + FNP group is compared to the positive control group, FNP. Body weight gain = Final body weight − Initial body weight. Gonadosomatic index = (Testes weight/body weight × 100).

**Table 4 toxics-11-00504-t004:** Oxidative stress, enzymatic and non-enzymatic antioxidants in rats treated with *Punica granatum* peel extract (PGPE), fenpropathrin (FNP), *Punica granatum* peel extract + fenpropathrin (PGPE + FNP).

Parameters	Experimental Groups			
	Control	PGPE	FNP	PGPE + FNP
TBARS (nmol/g tissue)	18.20 ± 0.725 ^c^	14.18 ± 0.591 ^d^	26.08 ± 0.616 ^a^	22.63 ± 0.434 ^b^
H_2_O_2_ (μmol/g tissue)	40.99 ± 0.663 ^c^	32.34 ± 0.707 ^d^	59.11 ± 1.639 ^a^	50.62 ± 1.982 ^b^
PCC (nmol carbonyl/mg protein)	0.41 ± 0.02 ^c^	0.42 ± 0.01 ^c^	1.36 ± 0.07 ^a^	0.68 ± 0.04 ^b^
SOD (U/mg protein)	69.07 ± 2.71 ^b^	84.50 ± 2.11 ^a^	35.49 ± 1.23 ^d^	48.14 ± 1.56 ^c^
CAT (U/mg protein)	7.13 ± 0.21 ^b^	8.47 ± 0.23 ^a^	3.74 ± 0.108 ^d^	5.17 ± 0.17 ^c^
GPx (U/mg protein)	7.59 ± 0.172 ^b^	8.94 ± 0.297 ^a^	4.226 ± 0.181 ^d^	5.774 ± 0.177 ^c^
GR (U/mg protein)	20.33 ± 0.492 ^b^	24.38 ± 0.591 ^a^	11.37 ± 0.338 ^d^	15.14 ± 0.543 ^c^
GST (µmol/hr/mg protein)	0.611 ± 0.024 ^b^	0.741 ± 0.022 ^a^	0.310 ± 0.013 ^d^	0.452 ± 0.019 ^c^
GSH (mmol/mg protein)	2.44 ± 0.067 ^b^	2.92 ± 0.084 ^a^	1.19 ± 0.054 ^d^	1.76 ± 0.067 ^c^

Values are expressed as means ± SEM; *n* = 7/group. Significance (*P* < 0.05) between groups was indicated in each row with different superscript letters (a,b,c,d). Statistically significant variations are compared as follows: the PGPE and FNP groups are compared to the negative control group, while the PGPE + FNP group is compared to the positive control group, FNP.

**Table 5 toxics-11-00504-t005:** The effect of PGPE, FNP and their combination on the activity of 3β-HSD, 17β-HSD, AST, ALT, ALP, and ACP, and protein content.

Parameters	Experimental Groups			
	Control	PGPE	FNP	PGPE + FNP
**Homogenate**				
3β-HSD (μmol/min/mg protein)	0.28 ± 0.02 ^a^	0.27 ± 0.01 ^a^	0.14 ± 0.01 ^c^	0.20 ± 0.02 ^b^
17β-HSD (μmol/min/mg protein)	0.22 ± 0.005 ^a^	0.23 ± 0.008 ^a^	0.11 ± 0.005 ^c^	0.18 ± 0.02 ^b^
Protein content (mg/g tissue)	73.19 ± 2.10 ^a^	79.90 ± 2.27 ^a^	42.52 ± 1.98 ^c^	53.68 ± 2.35 ^b^
**Serum**				
AST (U/l)	53.65 ± 1.58 ^c^	50.57 ± 1.68 ^c^	74.01 ± 2.47 ^a^	63.75 ± 1.85 ^b^
ALT (U/l)	56.77 ± 1.69 ^c^	53.90 ± 2.04 ^c^	78.80 ± 2.82 ^a^	69.13 ± 2.37 ^b^
ALP (U/l)	63.93 ± 2.21 ^c^	59.27 ± 2.00 ^c^	88.90 ± 3.19 ^a^	76.70 ± 2.29 ^b^
ACP (U/l)	10.42 ± 0.430 ^c^	9.45 ± 0.297 ^c^	14.14 ± 0.466 ^a^	12.89 ± 0.392 ^b^

Values are expressed as means ± SEM; *n* = 7/group. Significance (*P* < 0.05) between groups is indicated in each row with different superscript letters (a,b,c). Statistically significant variations are compared as follows: the PGPE and FNP groups are compared to the negative control group, while the PGPE + FNP group is compared to the positive control group, FNP. Abbreviations: PGPE = *Punica granatum* peel extract, FNP = fenpropathrin, PGPE + FNP = *Punica granatum* peel extract + fenpropathrin.

**Table 6 toxics-11-00504-t006:** Histological scores in the rat testes of the different groups: *Punica granatum* peel extract (PGPE), fenpropathrin (FNP), and *Punica granatum* peel extract + fenpropathrin (PGPE + FNP).

Parameters	Control	PGPE	FNP	PGPE + FNP
Disorganized seminiferous tubules	+	++	++++	++
Characterize spermatogenic cells - Hyperchromatic - Necrotic	+++ + +	+++ ++ +	+ +++ +++	+++ ++ ++
Necrotic spermatocytes	+	+	+++	++
Dilated lumen	−	+	+++	++
Dilated interstitial tissue	+	++	++++	+++
Infiltrating lymphocytes	-	+	+++	++
Spermatid	++	++	-	+

Mild +, moderate ++, marked +++, severe ++++.

## Data Availability

The original contributions presented in the study are included in the article; further inquiries can be directed to the corresponding author.

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
