# Peer review of "Punica granatum (Pomegranate) Peel Extract Pre-Treatment Alleviates Fenpropathrin-Induced Testicular Injury via Suppression of Oxidative Stress and Inflammation in Adult Male Rats"

_toxics, 2023, doi:10.3390/toxics11060504_

Round 1

Reviewer 1 Report

The work evaluated the protective effects of Punica granatum in alleviating the toxic effects of Fenpropathrin on the testis. The work is interesting and the results are well described. Some information are required:

- ministerial approval number

- check that the number of animals is suitable for the space, 7 are too many may be.

- more information of  all kits used for oxidative stress and inflammation

Author Response

Thank you so much for your valuable comments. 

1- ministerial approval number

Reply:

The approval number is not present because the Institutional Animal Care and Use Committee of Alexandria University is recently constructed and it is not present at the time of this experiment. Therefore, there is no number for this experiment.

2- check that the number of animals is suitable for the space, 7 are too many may be.

Reply:

Cage :  595 x 380 x 200 mm –floor area ~1820 cm2    (7 rats a 200 –300g)

Rats will need a larger cage, a minimum of 20 inches long x 14 inches wide x 24 inches high.  According to DIRECTIVE 2010/63/EU OF THE EUROPEAN PARLIAMENT AND OF THE COUNCIL of 22 September 2010 on the protection of animals used for scientific purposes.

So, 7 rats per cage are suitable.

3- more information of  all kits used for oxidative stress and inflammation

Reply:

Oxidative stress, enzymatic and non-enzymatic antioxidants are evaluated according to previous studies (reference methods) and these are mentioned in the manuscript. 

Concerning inflammation, more information was added and highlighted in the manuscript.  

Also, the introduction, results, and conclusion sections are improved and highlighted in the manuscript.

Thank you again and we appreciate your interest in our manuscript.            

Reviewer 2 Report

Environmental pollutants can cause toxic effects on the reproductive system. Therefore, the search for protective substances is of practical interest. Punica granatum peel extract is an affordable and effective substance that can be successfully used as a biologically active protective supplement. The study is well planned, including the determination of the composition of the extract, the determination of the quality of rat sperm, the quantitation of hormones, the morphological study of tissue, the activity of tissue enzymes, gene expression, and the level of cytokines. The introduction and discussion are very well written, and the materials and methods are also very well presented. I think that this study is of great scientific and practical importance. While reading the work, I had a few questions, the answers to which, it seems to me, will improve the presentation of this very interesting work.

Major

The main doubt I have is the conclusion on apoptosis. We need cytological confirmation of apoptosis for reliable statements. Another option is to calculate the Bax/Bcl protein ratio. In this paper, we see only the results of gene expression, not confirmed by protein studies. It seems to me that the conclusions on apoptosis should be more careful, and the word "apoptosis" would be better removed from the title.

Minor

1. In the introduction, please indicate exactly which parameters were improved by the use of Punica granatum peel extract.

2. There are a number of abbreviations that are not deciphered in the text or are deciphered too late. For example, HSD (line 36), LPO (line 382), ALT, AST, ALP, ACP. Please check all abbreviations.

3. Tables 3 and 4 - what do a, b, c, and d mean? It remains unclear to me which groups there is a significant difference between. Describe it more clearly, please.

4. Use “control” instead of “cont.” (Fig. 3, 4, Table 3).

5. Also provide a transcript of all abbreviations in the captions under the figures.

6. Please check for typos, subscripts (e.g. O2, Line 424).

7. In Fig. 1 mark the x-axis

8. In Figure 2 give not lines, but experimental points with an error.

9. What gene was used as an internal standard?

Author Response

Thank you for your valuable and important comments.

Major

The main doubt I have is the conclusion on apoptosis. We need cytological confirmation of apoptosis for reliable statements. Another option is to calculate the Bax/Bcl protein ratio. In this paper, we see only the results of gene expression, not confirmed by protein studies. It seems to me that the conclusions on apoptosis should be more careful, and the word "apoptosis" would be better removed from the title.

The word "apoptosis" is deleted from the title.

???????

Minor

  1. In the introduction, please indicate exactly which parameters were improved by the use of Punica granatum peel extract.

Reply:

The following sentences and references are added and highlighted as follows:

These components act by enhancing oxidative biomarkers, scavenging reactive oxygen species, and neutralizing them (Mo et al., 2022). 

Moreover, treatment with pomegranate peel and seed oil extracts enhanced liver functioning, reduced DNA fragmentation, caspase-3, and malondialdehyde levels, and increased enzymatic and nonenzymatic antioxidants due to diethylnitrosamine injection (Shaban et al., 2013).

References:

Mo Y, Ma J, Gao W, Zhang L, Li J, Li J and Zang J (2022) Pomegranate Peel as a Source of Bioactive Compounds: A Mini Review on Their Physiological Functions. Front. Nutr. 9:887113. doi: 10.3389/fnut.2022.887113

Nadia Z. Shaban, Mohamed A.L. El- Kersh, Fatma H. El-Rashidy, Noha H. Habashy. Protective role of Punica granatum (pomegranate) peel and seed oil extracts on diethylnitrosamine and phenobarbital-induced hepatic injury in male rats. Food Chemistry 141 (2013) 1587–1596  

2- There are a number of abbreviations that are not deciphered in the text or are deciphered too late. For example, HSD (line 36), LPO (line 382), ALT, AST, ALP, ACP. Please check all abbreviations.

Reply:

All abbreviations are added and highlighted in the manuscript

3- Tables 3 and 4 - what do a, b, c, and d mean? It remains unclear to me which groups there is a significant difference between. Describe it more clearly, please.

Reply:

Different letters (a,b,c,d) show significant differences at P < 0.05. PGPE and FNP groups are compared to the negative control group while PGPE + FNP group is compared to the positive control group, FNP. Abbreviations: PGPE=Punica granatum peel extract, FNP= Fenpropathrin, PGPE+FNP= Punica granatum peel extract + Fenpropathrin. FSH=Follicle stimulating hormone, LH=Luteinizing hormone.

Figure 4. Relative expression of p53, Caspase-3, Bcl-2, and cytokine levels (IL1β and IL10) in rats’ testes tissues of different groups. Data are presented as fold change (mean) ± SEM (n = 7/group). Different letters (a,b,c,d) show significant differences at P < 0.05. PGPE and FNP groups are compared to the negative control group while PGPE + FNP group is compared to the positive control group, FNP. Abbreviations: PGPE=Punica granatum peel extract, FNP= Fenpropathrin, PGPE+FNP= Punica granatum peel extract + Fenpropathrin.

4- Use “control” instead of “cont.” (Fig. 3, 4, Table 3).

Reply:

Cont. is replaced by control in (Fig. 3, 4, Table 3).

5- Also provide a transcript of all abbreviations in the captions under the figures.

Reply:

Abbreviations are added and highlighted

6- Please check for typos, subscripts (e.g. O2, Line 424).

Reply:

This is corrected and highlighted in the manuscript

7- In Fig. 1 mark the x-axis

Reply:

The x-axis represents the Time (min.) is added.

8- In Figure 2 give not lines, but experimental points with an error.

Reply:

This is corrected- Figure 2B

9- What gene was used as an internal standard?

Reply:

The following sentence is added and highlighted in the manuscript:

The β-actin gene was utilized as a reference (internal control) to calculate the fold change in target genes.

 Thank you again and we appreciate your interest in our manuscript.            

Reviewer 3 Report

Please see the marked manuscript for detailed comments to help with a revision.

Tables and figures need more information.  See suggestions.

Please make the Discussion more to the point.  Discuss your results in context of other findings.

Some editorial corrections would be good, but overall the English is fine.

Author Response

Thank you so much for your valuable comments.

1- Please see the marked manuscript for detailed comments to help with a revision.

Reply:

All detailed comments marked in the manuscript are considered. I hope it will be satisfactory.   

2-Tables and figures need more information.

Reply:

Abbreviations are added and highlighted within Tables and Figures.

Abbreviations: PGPE=Punica granatum peel extract, FNP= Fenpropathrin, PGPE+FNP= Punica granatum peel extract + Fenpropathrin. 

3- Please make the Discussion more to the point.  Discuss your results in context of other findings.

Reply:

Discussion is improved.

4- Some editorial corrections would be good, but overall the English is fine.

Reply:

English is revised 

Any corrections are highlighted in yellow color. 

Round 2

Reviewer 3 Report

The authors addressed many comments provided by this reviewer but some not addressed that must be are the following:

Include the common name of the plant in the title.

Line 27 - change cause to could.

Line 54 - insert visible before light

Line 108 - remove s from contents

Line 146 - remove s from sperms

Section 2.7.  Provide more info on CASA because this cannot be repeated.  What were the instrument settings to identify the cells?

Line 173 - remove s from phosphatases

Table 2.  Provide units of measure in PGPE column

Figs 1 and 2 are not helpful without more information written per figure.  This is not acceptable.

Line 259 - hormone levels

Lines 374-377 remain confusing as first noted in the previous review.  The authors need to address this.

Again, throughout the Discussion, the authors have not inserted references to their own results in terms of tables and figures.  They need to make an attempt at this to make their Discussion relevant.

Still could use help.  They are struggling.

Author Response

Response to Reviewer 3 comments - Round 2

Thank you so much for your valuable comments. Kindly find the reply to all comments step by step here and highlighted in blue color in the manuscript.

The authors addressed many comments provided by this reviewer but some not addressed that must be are the following:

1- Include the common name of the plant in the title.

Response 1:

Pomegranate is added in the title and highlighted.

2- Line 27 - change cause to could.

Response 2:

Cause is changed to could

3- Line 54 - insert visible before light

Response 3:

Visible is inserted before light

4- Line 108 - remove s from contents

Response 4:

S is removed

5- Line 146 - remove s from sperms

Response 5:

S is removed from sperms

6- Section 2.7.  Provide more info on CASA because this cannot be repeated.  What were the instrument settings to identify the cells?

Response 6:

The following part is added and highlighted:

Sperm samples were diluted with physiological solution (10 μl) and pipetted into a Makler Counting Chamber and immediately assessed. Within each of the measurements by the CASA system, parameters from a minimum of seven fields of Makler Counting Chamber were analyzed and 1000 sperms were evaluated per sample.

7- Line 173 - remove s from phosphatases

Response 7:

S is removed from phosphatases

8- Table 2.  Provide units of measure in PGPE column

Response 8:

units of measure are added in PGPE column

9- Figs 1 and 2 are not helpful without more information written per figure. This is not acceptable.

Response 9:

The following figure legends are added:

Figure 1. Representative GC-MS chromatogram of P. granatum peel extract and the fragmentation pattern of its constituents

(Retention time 30 min to 55 min).

Figure 2. Calibration curve of gallic acid (A) to determine total phenols content spectrophotometrically with different concentrations (0 to 20), catechin (B) to determine total flavonoids spectrophotometrically with different concentrations (0 to 400), and tannic acid (C) to determine total tannins spectrophotometrically with different concentrations (0 to 20). Each point represents the mean of three experiments.

10- Line 259 - hormone levels

Response 10:

Hormone levels is added

11- Lines 374-377 remain confusing as first noted in the previous review.  The authors need to address this.

Response 11:

This is addressed as follows:

Statistically significant variations are compared as follows: PGPE and FNP groups are compared to the negative control group while PGPE + FNP group is compared to the positive control group, FNP. Body weight gain= Final body weight- Initial body weight. Gonadosomatic index = (Testes weight/body weight × 100).

12- Again, throughout the Discussion, the authors have not inserted references to their own results in terms of tables and figures.  They need to make an attempt at this to make their Discussion relevant.

Response 12:

Discussion is improved and changes are highlighted.

Tables and Figures numbers are inserted and highlighted in the discussion

Thank you

Round 3

Reviewer 3 Report

The revisions have been adequately addressed.  The paper will be of interest to the readers.

Adequate